# Reduced Glx and GABA Inductions in the Anterior Cingulate Cortex and Caudate Nucleus Are Related to Impaired Control of Attention in Attention-Deficit/Hyperactivity Disorder

**DOI:** 10.3390/ijms23094677

**Published:** 2022-04-23

**Authors:** Ping C. Mamiya, Todd L. Richards, Richard A. E. Edden, Adrian K. C. Lee, Mark A. Stein, Patricia K. Kuhl

**Affiliations:** 1Institute for Learning & Brain Sciences, University of Washington, Seattle, WA 98195, USA; pkkuhl@uw.edu; 2Department of Radiology, University of Washington, Seattle, WA 98195, USA; toddr@uw.edu; 3Department of Radiology and Radiological Science, Johns Hopkins University, Baltimore, MD 21205, USA; richardedden@gmail.com; 4Department of Speech and Hearing Sciences, University of Washington, Seattle, WA 98195, USA; akclee@uw.edu; 5Department of Psychiatry and Behavioral Sciences, University of Washington, Seattle, WA 98195, USA; mark.stein@seattlechildrens.org

**Keywords:** neurotransmitter, anterior cingulate cortex, striatum, basal ganglia, executive function

## Abstract

Attention-deficit/hyperactivity disorder (ADHD) is a neurodevelopmental disorder that impairs the control of attention and behavioral inhibition in affected individuals. Recent genome-wide association findings have revealed an association between glutamate and GABA gene sets and ADHD symptoms. Consistently, people with ADHD show altered glutamate and GABA content in the brain circuitry that is important for attention control function. Yet, it remains unknown how glutamate and GABA content in the attention control circuitry change when people are controlling their attention, and whether these changes can predict impaired attention control in people with ADHD. To study these questions, we recruited 18 adults with ADHD (31–51 years) and 16 adults without ADHD (28–54 years). We studied glutamate + glutamine (Glx) and GABA content in the fronto-striatal circuitry while participants performed attention control tasks. We found that Glx and GABA concentrations at rest did not differ between participants with ADHD or without ADHD. However, while participants were performing the attention control tasks, participants with ADHD showed smaller Glx and GABA increases than participants without ADHD. Notably, smaller GABA increases in participants with ADHD significantly predicted their poor task performance. Together, these findings provide the first demonstration showing that attention control deficits in people with ADHD may be related to insufficient responses of the GABAergic system in the fronto-striatal circuitry.

## 1. Introduction

ADHD is a neurodevelopmental disorder that affects approximately 6% of children in the United States (http://cdc.gov/ncbddd/adhd/data.html (accessed on 7 March 2022)), and 5% of children worldwide [1]. Although ADHD symptoms may dissipate later in life, approximately two-thirds of affected individuals continue to have subthreshold symptoms and exhibit impairing symptoms in adulthood [2]. Thus, it is not surprising that 2.5% of adults have an ADHD diagnosis [3].

ADHD symptoms can vary throughout development. DSM-V provides three clinical presentations of ADHD: hyperactivity–impulsivity, inattentive, and combined. Children with ADHD often exhibit hyperactivity or impulsive behaviors [4,5,6,7]. As they enter into middle-school age, hyperactivity and impulsivity symptoms gradually decline and inattention gradually emerges and persists into adulthood, leading to lower educational attainment, disadvantageous socio-economic status, and long-term health crises [8,9,10,11].

Recent genome-wide association studies have identified genetic loci associated with ADHD etiology [9,12,13]. Among the reports, glutamate and GABA gene sets are associated with behavioral disinhibition in ADHD [13], suggesting that the imbalance of excitatory and inhibitory neurotransmitters may contribute to symptoms in ADHD.

Consistent with GWAS findings, studies using non-invasive magnetic resonance spectroscopy (MRS) analyses have revealed that glutamate and GABA content in the fronto-striatal circuitry are altered in people with ADHD. Notably, people with ADHD show altered glutamate and GABA content in the anterior cingulate cortex (ACC) and the striatum in the circuitry [14,15,16,17,18,19,20,21] (but see also [22,23].

However, emerging evidence has begun to show that glutamate and GABA content are dynamically regulated by environmental changes [24,25,26]. For example, recent evidence shows that glutamate content changes in the ACC when participants report various font colors in the Stroop task inside an MR scanner [27,28]. Moreover, greater glutamate changes were shown to be correlated with better task performance.

Although people with ADHD show altered glutamate and GABA content at rest in the fronto-striatal circuitry, it remains unknown whether people with ADHD show glutamate and GABA changes that are different from people without ADHD while performing attention control tasks. If so, we want to understand whether differences in Glx or GABA changes would help us to predict attention control deficits in people with ADHD.

To study these questions, we recruited 34 people with or without ADHD. We used MRS to quantify the glutamate and GABA content in the ACC and the striatum while participants performed three attention control tasks insides the MR scanner. Glutamate increases have been reported while healthy participants monitor fonts in various colors [28]. Therefore, we anticipated that Glx content would increase while our participants performed the attention control tasks. Moreover, glutamate is the precursor for GABA synthesis and GABA increases have been shown in participants performing a reinforcement learning task [29]. Therefore, we hypothesized that GABA content would also increase.

## 2. Results

### 2.1. Participants with ADHD Showed Smaller Glx Increases during the Tasks

A recent study by Taylor et al. (2015) has shown that glutamate content in the anterior cingulate cortex (ACC) increases when healthy human participants reported various font colors in the Stroop task [28]. Here, we asked whether same observations can be seen in people with ADHD. Consistent with the report by Taylor et al. (2015), we found that Glx content increased while participants performed the tasks (two-way repeated measures ANOVA: *F*_(3,126)_ = 9.820, *p* = 7.2 × 10^−6^, η_p_^2^ = 0.190). Notably, these increases were significantly smaller in the ADHD than the CONTROL group, indicated by a significant group effect on Glx increases (for CONTROL: mean ± SEM = 12.86 ± 2.09, CI = [8.68, 17.04]; for ADHD: mean ± SEM = 8.07 ± 1.54, CI = [5.01, 11.13]; *F*_(1,126)_ = 10.295, *p* = 0.002, Figure 1A). A post hoc analysis revealed significant Glx increases while participants performed the auditory task (adjusted *p* = 0.004, 95% CI = [2.934, 20.309]), the Stroop task (adjusted *p* = 2 × 10^−4^, 95% CI = [5.524, 22.899]), and the Flanker task (adjusted *p* = 5 × 10^−5^, 95% CI = [−6.774, 24.150]). Relatedly, we found that Glx content at rest did not differ between ADHD and CONTROL groups (Kruskal–Wallis test: *H*_(1)_ = 0.015, *p* = 0.904).

### 2.2. Participants with ADHD Showed Smaller GABA Increases during the Tasks

Next, we wanted to understand whether GABA content would show an increase during the tasks. There is evidence that GABA content shows rapid increases in the human anterior cingulate when people watch different visual stimuli [29,30]. Here, we found that only the CONTROL group showed GABA content increases (mean ± SEM = 2.23 ± 1.01, CI = [0.51, 5.52]); the ADHD group did not show increases in GABA content (mean ± SEM = 2.91 ± 1.23), CI = [0.08, 4.34]; *F*_(1,125)_ = 4.651, *p* = 0.033, η_p_^2^ = 0.036, Figure 1B). Relatedly, GABA content at rest did not differ between the ADHD and CONTROL groups (Kruskal–Wallis: *H*_(1)_ = 0.04, *p* = 0.38).

### 2.3. Lower Control of Attention in the ADHD Group

Next, we wanted to understand whether differences in Glx and GABA increases could contribute to behavioral changes in the attention control tasks. We found that participants with ADHD committed more errors when they were monitoring arrow directions in the Flanker task (in the congruent condition - CONTROL: mean ± SEM = 0.23 ± 0.17, ADHD: mean ± SEM = 1.58 ± 0.58; Kruskal–Wallis test: *H*_(1)_ = 4.062, *p* = 0.04; in the incongruent condition - CONTROL: mean ± SEM = 0.71 ± 0.35, ADHD: mean ± SEM = 2.12 ± 0.62; Kruskal–Wallis test: *H*_(1)_ = 3.114, *p* = 0.070; Figure 2).

### 2.4. Lower Glx or GABA Content at Rest Predicted Greater Glx or GABA Increases during the Tasks

To better understand if smaller or absent Glx and GABA increases may be related to Glx or GABA content at rest, we studied the Glx and GABA content when participants were not performing any tasks. We did not find Glx or GABA content at rest to be different between the ADHD and CONTROL groups (for Glx: Kruskal–Wallis test: *H*_(1)_ = 0.015, *p* = 0.904; for GABA: Kruskal–Wallis: *H*_(1)_ = 0.04, *p* = 0.38). However, we found that the amount of Glx and GABA at rest significantly predicted how great an increase would be seen during the tasks (for Glx in block2: *R*^2^ = 0.27, *p* = 0.004; block3: *R*^2^ = 0.41, *p* = 0.0001; block4: *R*^2^ = 0.22, *p* = 0.015; for GABA in block2: *R*^2^ = 0.25, *p* = 0.006; block3: *R*^2^ = 0.19, *p* = 0.027; block4: *R*^2^ = 0.24, *p* = 0.009). Participants with a lower Glx or GABA content at rest showed greater Glx or GABA increases (Figure 3).

### 2.5. Volumes of Gray-Matter, White-Matter, and Cerebropinal Fluid Were Identical between ADHD and CONTROL Groups

We performed an additional analysis to confirm that smaller Glx and GABA increases in the ADHD group were not due to anatomical difference in their brains. We did not find the volumes of gray matter (GM), white matter (WM), or cerebrospinal fluid (CSF) to be different between the ADHD and CONTROL groups (Appendix A). 

## 3. Discussion

Difficulties in inhibiting distractions and sustaining attention are hallmark symptoms of ADHD. Genome-wide association studies have revealed that glutamate and GABA gene sets are associated with ADHD symptoms. Consistently, brain imaging studies have provided evidence to show that glutamate + glutamine (Glx) and GABA content at rest in the fronto-striatal circuitry differ between people with or without ADHD. However, emerging evidence has suggested that glutamatergic and GABAergic systems are highly malleable to environmental changes or task demands. It remains unknown whether Glx and GABA content show changes when people are controlling their attention to conflicting presentations, and if so, whether these changes may explain the impaired control of attention in people with ADHD. To answer these questions, we studied Glx and GABA content when our participants performed attention control tasks. We found that participants with ADHD showed smaller Glx and GABA increases while performing the attention control tasks inside the MR scanner compared to participants without ADHD. Remarkably, participants with lower basal Glx or GABA content showed greater Glx or GABA increases during the tasks. Finally, we showed that participants’ performance in the task can be predicted by the GABA content during the task and by ADHD diagnosis. Together, our findings demonstrate that Glx and GABA content in the fronto-striatal circuitry exhibit increases when people direct their attention to conflicting visual presentations in tasks. These increases are smaller in people with ADHD, suggesting that impaired attention control may be attributed to insufficient GABA signaling in the fronto-striatal circuitry.

### 3.1. Reduced GABA and Glx Content during the Tasks Suggest Altered Glutamate-GABA Cycling in Subjects with ADHD

The present study used MEGA-PRESS GABA-editing pulses to reveal dynamic changes in Glx and GABA content in the ACC and the caudate nucleus during attention control tasks. The existing literature has shown that glutamate + glutamine (Glx) and GABA content at rest are altered in children and adults with ADHD [17,21]. Here, we furthered this understanding by showing that participants with ADHD showed smaller Glx and GABA increases compared to participants without ADHD (Figure 2B). These findings suggest that the reduction of GABA and Glx increases during these tasks may reflect a neurochemical abnormality in people with ADHD.

Reduced Glx and GABA content during the tasks may reflect changes occurring in various steps in glutamate–GABA cycling, such as during synthesis, metabolism, and clearance. Increases in Glx and GABA content during the tasks (Figure 2) can reflect increased metabolic activity and a higher glutamate–GABA cycling rate [31,32,33,34,35,36,37,38] that take place in the extracellular space, cytoplasm, and the mitochondria within the voxel [39,40,41,42].

Intracellular glutamate–glutamine cycling increases when metabolism increases in neurons [43]. Importantly, glutamate metabolism is implicated in ADHD. Polymorphisms in a gene catalyzing GABA synthesis (glutamate decarboxylase 1, *GAD1*) are associated with hyperactivity and impulsivity symptoms in ADHD [44,45]. The GAD enzyme in neurons catalyzes glutamate–GABA conversion [46]. There is evidence that GAD enzymatic activity highly depends on neural activity and that increased GAD activity causes increased GABA conversion in the striatum [24]. In the present study, we did not observe any differences in Glx or GABA concentrations during the non-task condition (block1) between subjects with and without ADHD. Thus, reduced GABA concentrations during the tasks may be related to reduced GAD activity, not a gross reduction in GABAergic neurons in the brain. Further investigations are warranted to characterize whether *GAD1* polymorphisms can affect GABA concentrations during these tasks in people with ADHD.

While altered metabolism in glutamate–GABA cycling may help explain decreased Glx and GABA concentrations in participants with ADHD, GABA and glutamate clearance in the synaptic clefts could also impact Glx and GABA quantification in the voxel. Once released from neurons, glutamate and GABA transporters uptake these metabolites from the synaptic clefts into astrocytes in order to maintain the homeostasis of excitatory and inhibitory neurotransmission in the local neural circuitry. Both GABA and glutamate transporters are strongly implicated in impaired cognition in neuropsychiatric disorders [13,47,48,49]. It is known that excessive glutamate in the synaptic cleft can cause neurotoxicity, leading to neuronal death [40]. Considering the role of GABA concentrations in sharpening the signal-to-noise ratio, it is not surprising that depletion of GABA transporters can cause hyperactivity and inattention phenotypes in animal models of ADHD [50]. Consistently, several single-nucleotide polymorphisms (SNPS) in the GABA transporter 1 (*GAT1)* gene are associated with symptoms in people with ADHD [51]. Thus, alterations in GABA and Glx clearance during tasks may be related to reduced GABA and Glx concentrations in participants with ADHD. Future investigations using animal models of ADHD are needed to verify the association between GABA and Glx transporter bindings and the MRS-based GABA and Glx concentrations.

### 3.2. The Inverse Relationship between Basal Glx or GABA Content and the Extent of Glx or GABA Increases May Be Related to Reduced GAD Activity

Glutamatergic and GABAergic systems are highly malleable to changes in the environment. In the mouse visual cortex, increasing neural activity results in increases in both neural excitation and inhibition [52]. Equalizing neural excitation with equivalent inhibition is thought to provide a pivotal role in maintaining the balance of neural networks. Here, we reported that basal Glx content was significantly correlated with basal GABA content. This relationship, however, was not significant when our participants performed the attention control tasks insides the MR scanner. The lack of relationship during the task stands in contrast with other reports that show a positive correlation when human participants face uncertain conditions in a behavioral paradigm, or when mice receive photoactivation in the layer 4 pyramidal cells of the visual cortex [29,52]. A positive correlation between Glx and GABA content is thought to be important for the balance of neural excitation and inhibition (E/I). Relatedly, disruptions in the E/I balance are linked to ADHD etiology [13]. However, GABA increases following stimulation have limitations. For example, the administration of a GABA transporter blocker increases GABA content up to a certain dose [53], but raising the dose of the GABA transporter blocker cannot further increase GABA content. This apparent “ceiling effect” on GABA increases may be related to decreased GAD67 enzymatic activity. GAD67 is a rate-limiting enzyme. An elevated synaptic GABA content may reduce GAD67 enzyme activity in the neurons. However, our findings do not allow the delineation of GABA content between the intracellular and extracellular space. It remains to be determined whether increased GABA content during the tasks is associated with synaptic GABA increases.

Notably, we found that greater GABA increases during the tasks were predicted by a lower basal GABA content. Cai et al. (2012) reported a similar finding that increases in GABA content after gabapentin administration were negatively correlated with GABA content before gabapentin administration [30]. This inverse relationship may be also related to GAD67 enzyme activity, which controls GABA synthesis depending on the synaptic GABA content. Further investigations are required to determine the relationship between GAD67 and GABA increases in mammalian brains.

### 3.3. Reduced GABA Increases during Tasks Contribute to Impaired Attention Control in ADHD

Our participants with ADHD showed smaller Glx and GABA increases during the tasks in which they showed impaired performance compared to participants without ADHD. Our findings add to the growing body of the literature that task-related GABA content predicts task performance [29,54,55,56]. Importantly, attention control deficits in ADHD are likely attributed to reduced GABA increases in the fronto-striatal circuitry.

Our group analysis revealed that participants with ADHD had significantly higher error rates in the Flanker task compared to participants without ADHD. It is worth noting that our participants with ADHD performed three attention control tasks with very minimal movements during the brain scan. Critically, they exhibited a significantly lower level of attention control, but not higher motion artifacts than the participants in the CONTROL group. These observations confirm the clinical view that adults with ADHD exhibit more inattentive or combined but less hyperactivity presentation. The trend of non-significantly higher rates in the Stroop task was unexpected given that behavioral studies have repeatedly shown that people with ADHD show impaired performance in this task [57,58,59,60]. This may be due to the limitations of the study. First, our attention tasks were presented in a pre-set order. Therefore, the order of the task presented during the scan may have contributed to task performance. In addition, our sample size was modest, allowing us to only provide a medium effect size.

## 4. Methods and Materials

### 4.1. Study Participants

The study consisted of 18 participants in the ADHD group and 16 participants in the CONTROL group (Appendix A). Participants in the ADHD group were recruited by sending the study flyer to patients in the “Mother First, Father Second” cohorts at the “Program to Enhance Attention, Regulation, and Learning” clinics at the Seattle Children’s Hospital [61]. Interested participants contacted researchers and were screened afterwards. Participants in the CONTROL group were recruited from the local community and were matched by gender, age, language, educational backgrounds, and employment status with participants in the ADHD group. Our participants with ADHD were carefully screened. Any individuals with a history of other mental disorders were not invited to participate in the study; this allowed us to eliminate the potential effects of other psychopharmacological treatments on the brain’s chemistry, including glutamate and GABA.

All participants met the following criteria: no history of other mental disorders, neurological impairments, developmental disorders, or hearing loss, and right-handed. The average age did not differ between ADHD (31–51 years) and CONTROL (28–54 years; *F*_(1,33)_ = 0.159, *p* = 0.692) groups. All experimental procedures were approved by the University of Washington Institutional Review Board and conformed to the ethical principles for research on human subjects from the Declaration of Helsinki, as revised in 2008. All participants gave written consent to participate in the study.

### 4.2. Experimental Design

All participants underwent a magnetization-prepared rapid gradient-echo (MPRAGE) T1-weighted imaging and magnetic resonance spectroscopy (MRS) scans and remained still during the T1 scan. MRS scans consisted of four eight-minute blocks (Figure 4A). In the first block, participants were not required to perform any tasks. In the second block, participants responded to either low- or high-pitched tones presented in the auditory task. In the third block, participants viewed various colors of fonts and performed the Stroop task. In the fourth block, participants viewed a set of conflicting arrows and performed the Flanker task. A magnetic resonance (MR) compatible button-box was placed in each hand.

### 4.3. Auditory Task

We adopted an auditory task that we previously published to assess participants’ selective attention to auditory stimuli [62]. We used E-Prime to present a 500 ms visual cue (up or down arrow) followed by two spoken digits (high or low pitch) that were played simultaneously during a trial (Figure 4B). The pitch of the spoken digits was shifted up and down using Praat software [63] to create competing tokens at 185 Hz ± 4.25 semitones. An arrow cue indicated which spoken digit the subject should be attending to. An up arrow indicated that the target is a high-pitch digit, and a down arrow indicated that the target is a low-pitch digit. Spoken digits were either one, two, three, or four. Subjects were instructed to press the left button in their left hand if they heard the word “one,” the right button in their left hand for the word “two,” the left button in their right hand for the word “three,” and the right button in their right hand for the word “four”. All subjects completed 70 trials in the task.

### 4.4. Color-Naming Version of the Stroop Task

We adopted the procedures for the Stroop task that we previously published [64], and used E-Prime to present one of four fonts, GREEN, RED, YELLOW, and BLUE in a mirror inside the scanner (Figure 4B). Participants viewed the color of a font that was consistent with the meaning conveyed by the font in the congruent condition or inconsistent with the meaning of the font in the incongruent condition. We used a single block design and intermixed the congruent and incongruent trials within the block. Participants were instructed to identify the color of a font in both conditions and use the button-box to respond. The corresponding button for the color red was the red button; for the color green, the green button; for the color yellow, the yellow button; and for the color blue, the blue button. All participants completed 186 trials in the task.

### 4.5. Flanker Task

We adopted the procedures for the Flanker task that we have previously published [65] and used E-Prime to display 5 arrows in a mirror inside the scanner (Figure 4B). Participants were required to identify whether the arrow in the center image pointed in the same direction (congruent condition) or in the opposite direction (incongruent condition) as the flanking arrows. We used a single block design and intermixed the congruent and incongruent trials within the block. The left button in the participants’ right hand would be pressed if the center arrow pointed to the left. The right button in the participants’ right hand would be pressed if the center arrow pointed to the right. All subjects completed 151 trials in the task.

Participants were instructed to respond as soon as a stimulus was presented. An invalid trial was marked if there was no button pressed two seconds after a presentation. E-Prime recorded the time when a presentation was shown and the time when a button was pressed in every trial. The latency represented the reaction time (RT) in a given trial.

### 4.6. Magnetic Resonance (MR) Data Acquisition

MR measures were acquired on a Philips 3T Achieva scanner version 5.18 using a 32-channel head coil (Figure 4C). MPRAGE T1-weighted images were acquired using the following parameters: TR = 11 ms, TE = 2.3 ms, flip angle = 8°, 256 slices covering the entire train, field of view = 230 × 230 mm^2^, matrix size = 328 × 320 mm^2^, reconstructed voxel size = 0.68 × 0.68 × 0.70 mm^3^ and was used to place a voxel in each individual’s brain to perform partial volume tissue correction in the subsequent data analysis. The total scan time was four minutes and one second.

GABA-edited MR spectra were acquired using the MEGA-PRESS method (TE = 68 ms, TR = 1500 ms, 1024 sampling points) with an editing pulse applied either at 1.9 ppm (ON) or at 7.5 ppm (OFF) [66,67]. Water suppression was achieved with variable-power radio frequency pulses with optimized relaxation delays (VAPOR) at the beginning of the MRS scan [68]. A brain voxel size of 50 (anterior–posterior) × 40 (right–left) × 45 (foot–head) mm^3^ encompassed the anterior cingulate cortex and the head of the caudate nucleus (Figure 4D,E). The assessment of GABA with MEGA-PRESS included the co-editing of macromolecules, which contributed to the edited peak at 3 parts per million (ppm). Data quality was closely monitored and the acquisition was terminated and restarted if any movement occurred.

### 4.7. MRS Data Analyses

We used the Gannet3 toolbox to quantify Glx and GABA concentrations within the voxel [69]. Processing included automatic frequency and phase correction, artifact rejection (frequency correction parameters >3 SD above mean), 3 Hz exponential line broadening, and fitting of the creatine signals. After subtracting OFF from ON acquisitions, a single GABA peak at 3 ppm and a double Glx peak at 3.75 ppm were separately fitted using a five-parameter Gaussian mode. A GABA peak was fit with a Gaussian and the integral of the fit served as the concentration measurement. This GABA value was scaled by the integral of the unsuppressed water peak, fit with a mixed Gaussian–Lorentzian curve (Figure 4F). The surface area in a Glx peak in the MEGA-PRESS difference spectrum was estimated using a double Gaussian fit and normalized to water. An MRS voxel for each subject was co-registered to its respective structural image using GannetCoRegister [70]. This produced a binary voxel mask, which was segmented into gray matter, white matter, and cerebrospinal fluid (CSF) probabilistic partial volume maps using the unified tissue segmentation algorithm in SPM12 [71] provided by GannetSegment [70]. The GABA and Glx values were then corrected based on segmented T1-weighted images for each individual. To assess the consistency of the data quality, the full-width half-maximum (FWHM) of creatine and the fit errors of the GABA and Glx peaks were additionally assessed and only spectra with an FWHM of 20 Hz or less, or a fit error of less than 10%, were included in the analysis. We found that one subject with ADHD showed excessive motion during block3 and block4. The Glx and GABA model could not be fit in these blocks. Thus, the MRS data from this subject was not entered in the analysis.

We used the MEGA-PRESS OFF spectra and examined creatine (Cr), N-acetylaspartate (NAA), and N-acetylaspartylGlu (NAAG) signals using the LC Model (version 6.3-1L) [72]. Only subjects whose Cramer–Rao lower bounds (CRLB) were 20% or lower were entered in the analysis. We used the creatine signal to reference the spectral quality. We used creatine signals to understand the spectral quality and confirmed that the signals were consistent throughout the scan (repeated measures ANOVA: *F*_(3,128)_ = 0.042, *p* = 0.990; Appendix A). Similarly, the fit errors of Glx and GABA peaks also were consistent throughout the scan (for Glx: repeated measures ANOVA: *F*_(3,128)_ = 0.516, *p* = 0.672: for GABA: repeated measures ANOVAs: *F*_(3,128)_ = 1.207, *p* = 0.310; Appendix A).

### 4.8. Statistical Analyses

We use the tidyverse (version 1.3.0) and rstatix libraries in *R* (version 3.6.2) for data analysis. We calculated the average, standard deviation, and 95% confidence interval of measurement precision of Glx and GABA content (Appendix A). To test the hypothesis that the Glx content increased during tasks, we used the ‘group’ (ADHD versus CONTROL) as the between-subject factor, the ‘block’ (block1–block4) as the within-subject factor, and Glx changes (ΔGlx) as a dependent variable in a repeated measures analysis of variance (ANOVA). We added age as a covariate in the repeated measures analysis of covariance (ANCOVA)—because the age factor has been shown to greatly impact cortical GABA content [73,74]—and used group as a between-subject factor, block as a within-subject factor, and GABA changes (ΔGABA) as a dependent variable. We used quantile–quantile plots of ANOVA residuals to assess the normality of distribution. The control group was used as the reference level in the ‘group’ factor. Block 1 was used as the reference level in the ‘block’ factor. We used two-way repeated measure ANOVAs to assess the FWHM of Creatine and the fit errors of the Glx and GABA peaks and entered the group as the between-subject factor, block as the within-subject factor, and FWHM or fit error as a dependent variable. We used the non-parametric Kruskal–Wallis test to assess tissue components within the MRS voxel and the Glx and GABA content in the non-task condition (block1); Tukey’s Honest Significant Differences (HSD) was used as a post hoc test after ANOVA and adjusted the *p* value according to the number of comparisons.

We calculated task error rates by dividing the number of incorrect responses by the total number of trials in the congruent and incongruent condition, respectively. We found that one participant without ADHD and one participant with ADHD showed an error rate three SDs higher than their group average. Thus, their data were removed from the Stroop analysis. We used the non-parametric Kruskal–Wallis test to understand behavioral measures in the tasks. We applied the Bonferroni correction and multiplied the resulting *p* values by two to account for two conditions in the task.

We used linear regression and entered Glx, GABA content, and group (ADHD vs. CONTROL) as predictors in a linear regression model. We used ANOVA to understand the model fit and the eta squared to determine the effect size of a predictor. We performed permutation tests with 500 iterations to confirm if the chance of obtaining the observed R-squared and *p* values of each model was greater than would be expected by chance (*p* < 0.05).

We used the Pearson correlation test to assess the relationship between the basal Glx/GABA content in the non-task condition and Glx/GABA content during the task, as well as between GABA and Glx content in the non-task and task conditions; we applied Bonferroni corrections to the resulting *p* values. We used an alpha level of 0.05 to determine statistical significance in the study.

## 5. Conclusions

We combined behavioral assessments of attention control with MRS imaging to demonstrate dynamic changes of Glx and GABA content in the ACC and caudate nucleus while participants performed three attention control tasks. We found that participants with ADHD showed smaller Glx and GABA increases during the tasks compared to participants without ADHD. We also found that basal Glx or GABA content predicted the extent of Glx or GABA increases during the tasks. Finally, GABA content during the task and ADHD diagnosis allowed us to predict the task performance. These findings demonstrate the dynamic role of glutamate–GABA cycling in the fronto-striatal circuitry and suggest that a smaller GABA content in the ACC and the caudate nucleus may contribute to impaired cognition in ADHD.

## Figures and Tables

**Figure 1 ijms-23-04677-f001:**
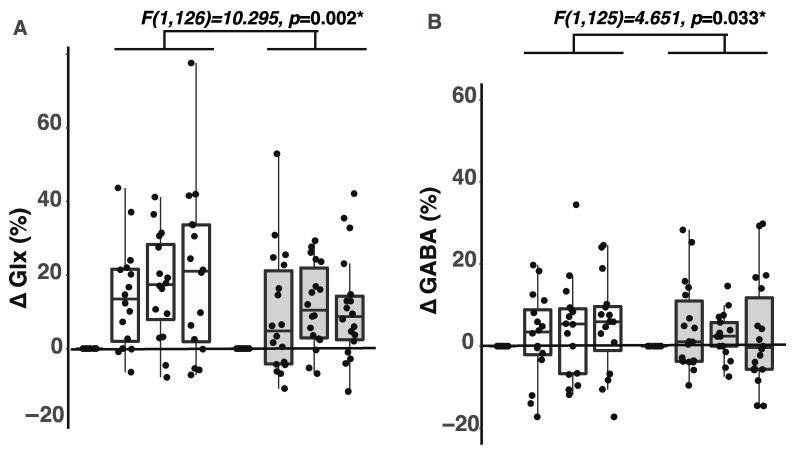
Task-related changes in Glx and GABA content. Open white bars represent the CONTROL group and grey bars represent the ADHD group. In each group, three bars are shown as ΔGlx or ΔGABA increases in the auditory, Stroop, and Flanker tasks respectively. (**A**) Glx changes during the tasks (ΔGlx) are shown in percentage. ΔGlx were computed using the following formula: ΔGlx = (Glx_task_ − Glx_non-task_)/Glx_non-task_ × 100%. (**B**) GABA changes during the tasks (ΔGABA) are shown in percentage. ΔGABA were computed using the following formula: ΔGABA = (GABA_task_ − GABA_non-task_)/GABA_non-task_ × 100%. In both figures, dots represent the data points from individual subjects. The upper boundary of an individual box represents the 75th percentile and the lower boundary represents the 25th percentile of the value for an individual block. The horizontal line within the box represents the median in a respective block. Asterisks represent significant changes in Glx or GABA content between ADHD and CONTROL groups at the *p* level of 0.05.

**Figure 2 ijms-23-04677-f002:**
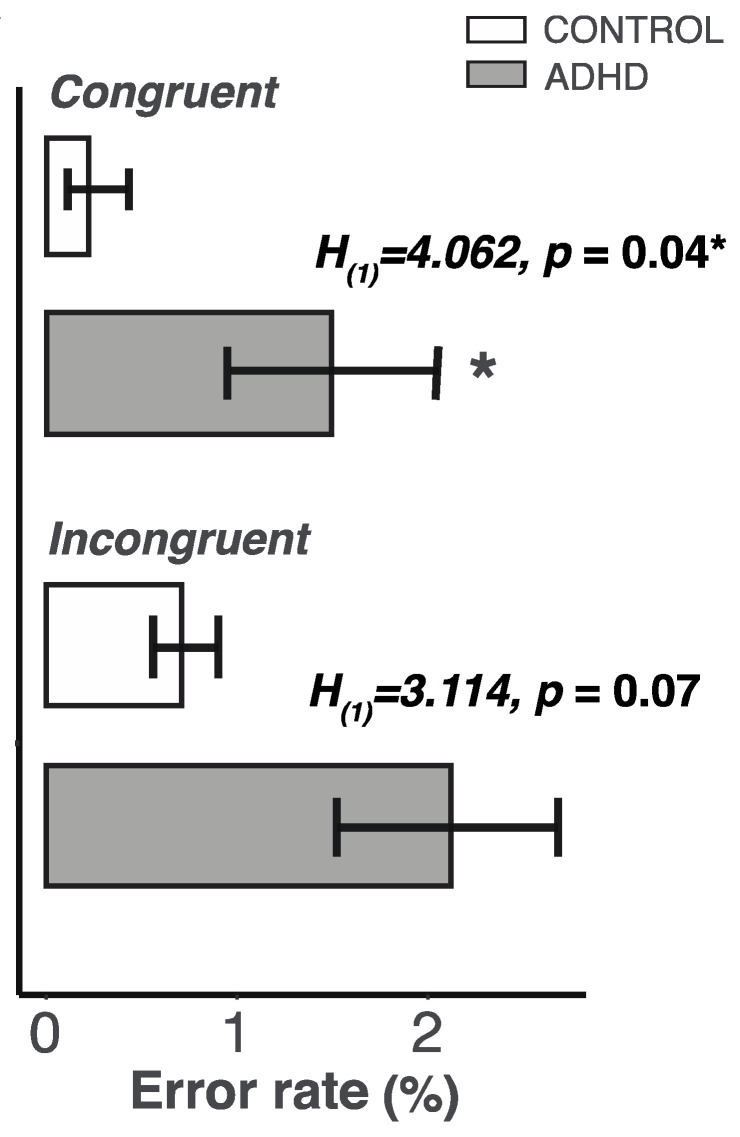
Participants with ADHD showed higher rates when they faced arrows pointing in the same direction (congruent condition). Open bars represent participants without ADHD and grey bars represent subjects with ADHD. Data is presented as mean with standard error bar. An asterisk indicates a significant higher error rate in ADHD than CONTROL group at a *p* level of 0.05.

**Figure 3 ijms-23-04677-f003:**
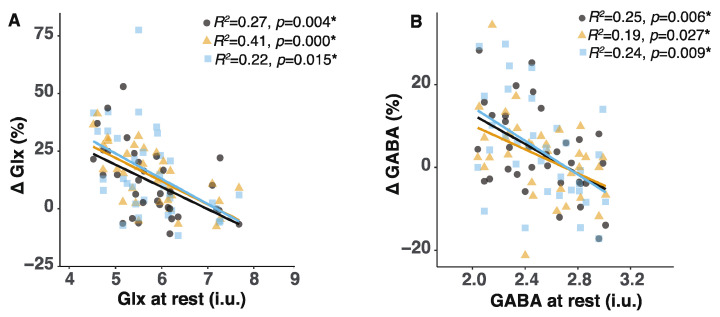
The degree of Glx and GABA increases during the tasks was predicted by the basal level of Glx and GABA content at rest. (**A**) Greater increases in Glx content during the tasks were predicted by a smaller Glx content at rest. (**B**) Greater increases in GABA content during the tasks were predicted by a smaller GABA content at rest. Gray circle symbols represent Glx or GABA increases in block2. Yellow triangle symbols represent Glx or GABA increases in block3. Blue square symbols represent Glx or GABA increases in block4. Asterisks represent that the Glx or GABA content at rest significantly predicted Glx or GABA increases during the tasks at the *p* level of 0.05.

**Figure 4 ijms-23-04677-f004:**
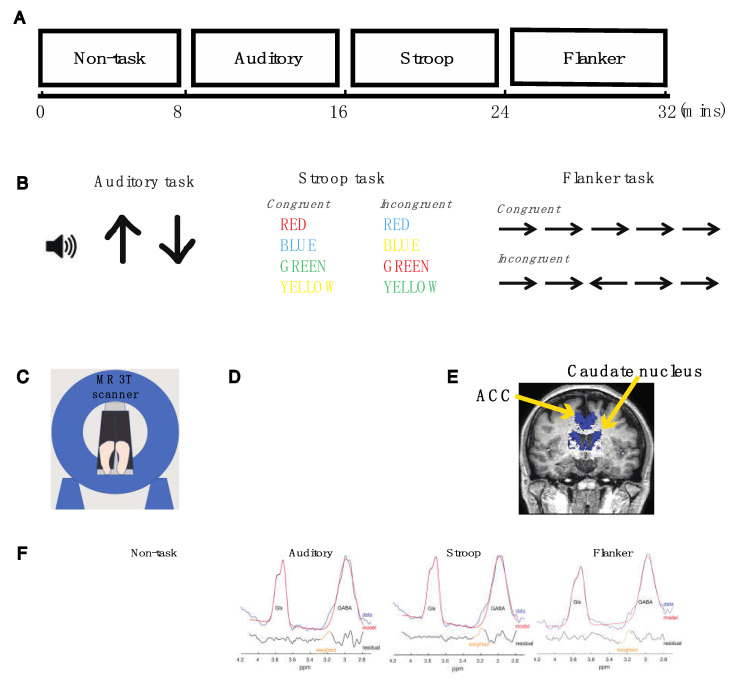
The experimental design, the voxel placement, and the quantification of Glx and GABA concentrations in the MRS experiment. (**A**) The entire MRS scan was divided into four blocks of time. Each block lasted eight minutes. In the 1st block, subjects were not required to perform any tasks and remained still insides the scanner. In the subsequent blocks, subjects performed auditory, Stroop, and Flanker tasks. (**B**) Three attention control tasks were administered during the MRS scan: study participants listened to two competing tones in the auditory task; study participants reported the colors of various fonts in the Stroop task; study participants reported the arrow direction of the center image in the Flanker task. (**C**) An illustration of a participant laying still inside a Philips Achieva 3T scanner. (**D**) The voxel placement overlaid on a subject’s T1-weighted image shown in the axial plane. (**E**) The anterior cingulate cortex (ACC) and the caudate nucleus within the MRS voxels shown in a T1-weighted image in the coronal plane. (**F**) Representative fits of a single GABA peak at 3 parts per million (ppm) and a double Glx peak at 3.75 ppm in the edited MEGA-PRESS spectrum during the non-task and task conditions. The red trace is the model fit overlaid on the raw MRS data.

## Data Availability

Data and the *R* codes are available upon reasonable requests.

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
