# Peer review of "Reduced Glx and GABA Inductions in the Anterior Cingulate Cortex and Caudate Nucleus Are Related to Impaired Control of Attention in Attention-Deficit/Hyperactivity Disorder"

_ijms, 2022, doi:10.3390/ijms23094677_

Round 1

Reviewer 1 Report

This is an interesting study showing the correlation between the induction of Glx and GABA activities in the  in the anterior cingulate cor- 2
tex and caudate nucleus and the performance of attention control tasks for ADHD patients. 

There are a few issues that need to be addressed:

1) The title of the manuscript is not straightforward and clear. Maybe considering changing it to 'Reduced Glx and GABA induction in the ...'?

2) Figure 2 is missing appropriate labeling and indication of the bars, colors and groups. 

3) A previous study by Puts et al. 2020 found a reduced striatal level of GABA in unmedicated children with ADHD, of which the conclusion is a bit different from the findings here (the authors claim a higher level of GABA/Glx at basel levels is correlated with less induction of GABA/Glx during task controlling). Could the authors comment whether is the age difference between the two studies that leads to the discrepancy or the brain region analyzed?

4) Since it's a correlational study, can the authors really draw the conclusion that attention control deficits in ADHD patients may attribute to insufficient responsiveness of GABA? (Line 26-28).

5) Did the authors analyze other brain regions?

6) There are a few typos and grammar mistakes that need to be improved. e.g. Line 312 'glutamate+glutamate (Glx)', Line 316 '...when people are control their attention...'

Author Response

Reviewer 1

Comments and Suggestions for Authors

This is an interesting study showing the correlation between the induction of Glx and GABA activities in the  in the anterior cingulate cortex and caudate nucleus and the performance of attention control tasks for ADHD patients. 

There are a few issues that need to be addressed:

  • The title of the manuscript is not straightforward and clear. Maybe considering changing it to 'Reduced Glx and GABA induction in the ...'?

Response: We would like to thank the reviewer for the suggestion. We have revised the title of our manuscript. The title now reads:

 “Reduced Glx and GABA inductions in the anterior cingulate cortex and caudate nucleus are related to impaired control of attention in attention-deficit/hyperactivity disorder”.

  • Figure 2 is missing appropriate labeling and indication of the bars, colors and groups. 

Response: We would like to thank the reviewer for pointing out the incomplete labelling in Figure 2. To improve the clarity, we add the following sentence:

“Open white bars represent the CONTROL group and the gray bars represent the ADHD group. In each group, three bars are shown as deltaGlx or deltaGABA increases in the auditory, Stroop, and Flanker tasks respectively.”

  • A previous study by Puts et al. 2020 found a reduced striatal level of GABA in unmedicated children with ADHD, of which the conclusion is a bit different from the findings here (the authors claim a higher level of GABA/Glx at basal levels is correlated with less induction of GABA/Glx during task controlling). Could the authors comment whether is the age difference between the two studies that leads to the discrepancy or the brain region analyzed?

Response. We would like to thank the reviewer for the question. We would like to clarify that there are differences between the present study and Puts et al (2020) in terms of the age of participants, voxel placement, and the experimental design. As the reviewer pointed out, participants in Puts et al.(2020) were children between 5-9 years of age. In the present study, the age range was between 31-51 years of age for the ADHD group. Next, Puts et al (2020) reported GABA content only when children were at rest insides the MR scanner. The authors reported that GABA content in the striatum was significantly less in the ADHD group than the CONTROL group, but no difference between groups in the anterior cingulate cortex (ACC). Interestingly, the authors also reported that participant’s age was positively correlated with the GABA content in the striatum. Older children had higher GABA content in the striatum. In the present study, we did not find GABA content in the ACC and caudate nucleus (part of the striatum) at rest to be different between ADHD and CONTROL groups. One plausible explanation for the difference between the present study and Puts et al (2020) is likely due to the age of participants. Children in Puts et al.(2020) may have higher striatal GABA content in the adulthood as GABA content increases over time. Or our participants may have had lower GABA content in the caudate nucleus when they were a child.  

4) Since it's a correlational study, can the authors really draw the conclusion that attention control deficits in ADHD patients may attribute to insufficient responsiveness of GABA? (Line 26-28).

Response: We would like to thank the reviewer for the suggestion. We have revised the sentence to emphasize the relationship, not the causal effect of GABA on attention control deficits in ADHD. In line 41, the new sentence now reads:

Together, these findings provide the first demonstration to show that attention control deficits in people with ADHD may be related to insufficient responses of the GABAergic system in the fronto-striatal circuitry.”

5) Did the authors analyze other brain regions?

Response: We would like to thank the reviewer for asking this question. We used single voxel MRS protocol and the within-subject experimental design that allowed us to assess the Glx and GABA increases during the tasks. This method has been shown to give higher precision in Glx and GABA quantification. However, the total brain scan time was 32 minutes. We did not acquire additional voxel in other brain regions out of the concerns that participants may become bored and start moving around during the scan. The potential motion artifacts will inevitably damage the data quality and affect our analysis. We hope that the reviewer will agree with our approach.

6) There are a few typos and grammar mistakes that need to be improved. e.g. Line 312 'glutamate+glutamate (Glx)', Line 316 '...when people are control their attention...'

Response: We would like to thank the reviewers for pointing out our grammar errors. We revised the sentences in both places. The new sentences now read:

In line 382, “Consistently, brain imaging studies have provided evidence to show that glutamate + glutamine (Glx) and GABA content at rest in the fronto-striatal circuitry differ between people with or without ADHD.

In line 386, “It remains unknown whether Glx and GABA content would show changes when people are controlling their attention to conflicting presentations”.

Reviewer 2 Report

This is a very interesting paper.

The abstract should be a more structured form.

In the introduction should be a more detailed clinical description of ADHD (symptoms, variants, etc.) Differents between clinical symptoms in different ages should be discussed.

line 59. Abbreviation MRS used without explanation, which is in line 86 (the same problem with other abbreviations ex. MR)

The participants' age is high which implicates specific symptoms and variants of ADHD - it should be discussed.

The medications in patients with ADHD should be better described and discussed. In supplementary materials, I found information:

Methylphenidate: 1
Dextroamphetamine: 2
Ritalin: 1
No information about doses.

What is the difference between  Methylphenidate and Ritalin?

Can you have any idea if the dopaminergic medications can change your results?

Why only 4 patients is on pharmacological treatment?
It is unusual. Adult patients with ADHD without pharmacological treatment more frequently have other psychopathological symptoms it should be discussed. 

line 422. "We" not "we"

Author Response

Comments and Suggestions for Authors

This is a very interesting paper.

  • The abstract should be a more structured form.

Response: We would like to thank the reviewer for the comment. We have revised the abstract to make it more structured and clear. In line 27, the revised abstract now reads:

Attention-deficit/hyperactivity disorder (ADHD) is a neurodevelopmental disorder that impairs the control of attention and behavioral inhibition in affected individuals. Recent genome-wide association findings reveal an association between glutamate and GABA gene sets and ADHD symptoms. Consistently, people with ADHD show altered glutamate and GABA content in the brain circuitry that is important for attention control function. Yet, it remains unknown how glutamate and GABA content in the attention control circuitry would change when people are controlling their attention, and whether these changes would predict impaired attention control in people with ADHD. To study these questions, we recruited 18 adults with ADHD (31-51 years) and 16 adults without ADHD (28-54 years). We studied glutamate+glutamine (Glx) and GABA content in the fronto-striatal circuitry while participants performed the attention control tasks. We found that Glx and GABA concentrations at rest did not differ between participants with ADHD or without ADHD. However, while participants were performing the attention control tasks, participants with ADHD showed smaller Glx and GABA increases than participants without ADHD. Notably, smaller GABA increases in participants with ADHD significantly predicted their poor task erformance. Together, these findings provide the first demonstration to show that attention control deficits in people with ADHD may be related to insufficient responses of the GABAergic system in the fronto-striatal circuitry.”

  • In the introduction should be a more detailed clinical description of ADHD (symptoms, variants, etc.) Differences between clinical symptoms in different ages should be discussed.

Response: We would like to thank the reviewer for the suggestion. We have added new information to discuss the clinical presentations of ADHD throughout development. In line 83, new sentences now read:

ADHD is a neurodevelopmental disorder that affects approximately 6% of children in the United States (http://cdc.gov/ncbddd/adhd/data.html), and 5% of children worldwide (1). Although ADHD symptoms may dissipate later in life, approximately two-thirds of affected individuals continue to have subthreshold symptoms and exhibit impairing symptoms in the adulthood (2). Thus, it is not surprising that 2.5% of adults have ADHD diagnosis (3).

ADHD symptoms can vary throughout development. DSM-V provides three clinical presentations of ADHD: hyperactivity – impulsivity, inattentive, and combined. Children with ADHD often exhibit hyperactivity or impulsive behaviors (4-7). As they enter the middle-school age, hyperactivity and impulsivity symptoms gradually decline and inattention gradually emerge and persist into adulthood, leading to lower educational attainment, disadvantageous socio-economic status, and long-term health crisis (8-11).”

  • line 59. Abbreviation MRS used without explanation, which is in line 86 (the same problem with other abbreviations ex. MR).

Response: We have provided the full name of magnetic resonance spectroscopy (MRS) in the Introduction. In the third paragraph in the Introduction, the first sentence reads: “Consistent with GWAS findings, studies using non-invasive magnetic resonance spectroscopy (MRS) analyses have revealed that glutamate and GABA content in the fronto-striatal circuitry are altered in people with ADHD.”

To further facilitate readers’ understanding, we added the full name of MRS and MR in the experimental design in the Methods and Materials section in this revision. The new sentences now read:

All participants underwent a magnetization-prepared rapid gradient-echo (MPRAGE) T1-weighted imaging and magnetic resonance spectroscopy (MRS) scans and remained still during the T1 scan.”

A magnetic resonance (MR) compatible button-box was placed in each hand.

4) The participants' age is high which implicates specific symptoms and variants of ADHD - it should be discussed.

Response: We would like to thank the reviewer for suggesting further discussion about symptoms and variations of ADHD in our manuscript. We have added new sentences to discuss this. In line 483, new sentences now read:

It is worth noting that our participants with ADHD performed three attention control tasks with very minimal movements during the brain scan. Critically, they exhibited significantly lower level of attention control but not higher motion artifacts than the participants in the CONTROL group. These observations conform the clinical view that adults with ADHD exhibit more inattentive or combined but less hyperactivity presentation.”

  • The medications in patients with ADHD should be better described and discussed. In supplementary materials, I found information:

Methylphenidate: 1
Dextroamphetamine: 2
Ritalin: 1
No information about doses.

What is the difference between  Methylphenidate and Ritalin?

  • Response: One participant who reported Methylphenidate has taken the generic methylphenidate medication, such as Methylphenidylacetate hydrochloride, whereas another participant reported to take the brand name, Ritalin for the treatments.
  • Can you have any idea if the dopaminergic medications can change your results?

Response: Recent literature has shown that dopaminergic medications do not affect Glx or GABA content in the medial prefrontal cortex or the anterior Cingular cortex in adults with ADHD. This is uniquely different from children with ADHD whose glutamate+glutamine (Glx) or GABA content is more likely to be affected by Methylphenidate treatment. These observations suggest that dopaminergic treatments are more effective in changing neuro-metabolites during the early stage of life.

  • Why only 4 patients is on pharmacological treatment?
    It is unusual. Adult patients with ADHD without pharmacological treatment more frequently have other psychopathological symptoms it should be discussed. 

Response: We would like to thank the reviewer for the suggestion. Our participants in the ADHD group who did not take pharmacological treatments for various reasons, such as lose of insurance coverage, financial burden, or use behavioral therapy. To control for the effects of any psychiatric or mental disorders, we have carefully screened all participants in both control and ADHD groups to ensure that no participant has had any history of these diagnoses. This information has been provided in the Methods and Materials Section, under the Study Participants. To emphasize our screening efforts, we now added a new sentence in the same section. In line 93 on page 4, the new sentences now read::

“Our participants with ADHD have been carefully screened. Any individuals with a history of other mental disorders were not invited to participate in the study. This allows us to eliminate the effects of other psychopharmacological treatments might have had on the brain chemistry, including glutamate and GABA.”

  • line 422. "We" not "we"

Response: We would like to thank the reviewer for pointing out our error. We revised the spelling. The new sentence now reads:

“We combined behavioral assessments of attention control with MRS imaging to demonstrate the dynamic changes of Glx and GABA content in the ACC and caudate nucleus while participants performed three attention control tasks.”

Round 2

Reviewer 2 Report

accept in current form